META-RESEARCH ARTICLE

# The manifold costs of being a non-native English speaker in science

Tatsuya Amano[1,2]*, Valeria Ramírez-Castañeda[3,4], Violeta Berdejo-Espinola[1,2], Israel Borokini[5], Shawan Chowdhury[1,2,6,7,8], Marina Golivets[9], Juan David González-Trujillo[10], Flavia Montaño-Centellas[11,12], Kumar Paudel[13], Rachel Louise White[14], Diogo Veríssimo[15]

**1** School of the Environment, The University of Queensland, Brisbane, Queensland, Australia, **2** Centre for Biodiversity and Conservation Science, The University of Queensland, Brisbane, Queensland, Australia, **3** Museum of Vertebrate Zoology, University of California, Berkeley, California, United States of America, **4** Department of Integrative Biology, University of California, Berkeley, California, United States of America, **5** University and Jepson Herbaria, Department of Integrative Biology, University of California, Berkeley, California, United States of America, **6** Institute of Biodiversity, Friedrich Schiller University Jena, Jena, Germany, **7** Department of Ecosystem Services, Helmholtz Centre for Environmental Research - UFZ, Leipzig, Germany, **8** German Centre for Integrative Biodiversity Research (iDiv) Halle-Jena-Leipzig, Leipzig, Germany, **9** Department of Community Ecology, Helmholtz Centre for Environmental Research - UFZ, Halle, Germany, **10** Museo Nacional de Ciencias Naturales (CSIC-MNCN), Madrid, Spain, **11** Instituto de Ecología, Universidad Mayor de San Andrés, La Paz, Bolivia, **12** Department of Biological Sciences, Louisiana State University, Baton Rouge, Louisiana, United States of America, **13** Greenhood Nepal, Kathmandu, Nepal, **14** School of Applied Sciences, University of Brighton, Brighton, United Kingdom, **15** Department of Biology, University of Oxford, Oxford, United Kingdom

* t.amano@uq.edu.au

**Data Availability Statement:** The data underlying Figs 2B, 3A, 3B, 4C, S1, S2, S4, S5, S6, S7, S8, and S9 can be found in S1 Data. The data underlying Figs 1A, 1B, 1C, 1D, 1E, 1F, 2A, 2C, 2D, 4A, 4B, and

## Abstract

The use of English as the common language of science represents a major impediment to maximising the contribution of non-native English speakers to science. Yet few studies have quantified the consequences of language barriers on the career development of researchers who are non-native English speakers. By surveying 908 researchers in environmental sciences, this study estimates and compares the amount of effort required to conduct scientific activities in English between researchers from different countries and, thus, different linguistic and economic backgrounds. Our survey demonstrates that non-native English speakers, especially early in their careers, spend more effort than native English speakers in conducting scientific activities, from reading and writing papers and preparing presentations in English, to disseminating research in multiple languages. Language barriers can also cause them not to attend, or give oral presentations at, international conferences conducted in English. We urge scientific communities to recognise and tackle these disadvantages to release the untapped potential of non-native English speakers in science. This study also proposes potential solutions that can be implemented today by individuals, institutions, journals, funders, and conferences.

Please see the Supporting information files (S2–S6 Text) for Alternative Language Abstracts and Figs 5 and 6.

S3 are raw data directly from the survey questions, which our ethics approval prevents us from sharing to secure confidentiality of the respondents. Access to these raw data should be requested from the University of Queensland Ethics office, which can be contacted at humanethics@research.uq.edu.au. All codes used in the analysis are available at: http://doi.org/10.17605/OSF.IO/Y94ZT.

**Funding:** This work was funded by the Australian Research Council Future Fellowship FT180100354 (TA), The University of Queensland strategic funding (TA), and the German Research Foundation (DFG-FZT 118, 202548816) (SC). The funders had no role in study design, data collection and analysis, decision to publish, or preparation of the manuscript.

**Competing interests:** The authors have declared that no competing interests exist.

## Introduction

Unlocking the potential of disadvantaged communities is one of the urgent challenges in science. Collaboration involving a diverse group of people can better solve problems [1] and deliver higher levels of, and more relevant, scientific innovation [2] and impacts [3]. Today, the need to tap into a diversity of people, views, knowledge systems, and solutions in order to successfully address global challenges, such as the biodiversity and climate crises [4–6], is being increasingly recognised, and there is a critical need to do so across multiple disciplines [7–9].

Increasing the diversity within scientific communities requires breaking down the barriers that impede the career development of disadvantaged groups of researchers, and one such barrier is rooted in language. Although the use of English as the common language of science has no doubt contributed to the advance of science [10], this benefit comes with considerable costs for those whose first language is not English (hereafter, non-native English speakers). Non-native English speakers, who constitute the majority of the world's population, face a number of challenges in conducting and communicating science in English, which inevitably impose an excessive burden on their career development in science. This issue is widely recognised [11,12], as English now plays a dominant role in the execution and communication of science, as well as the evaluation of scientists, in almost any scientific discipline [13]. For example, the United Nations Educational, Scientific, and Cultural Organization (UNESCO)'s recommendation on open science, adopted by 193 member states in 2021, highlights the need to overcome language barriers in order to achieve 4 of the open science core values and guiding principles (Equity and fairness, Diversity and inclusiveness, Equality of opportunities, and Collaboration, participation and inclusion) [14]. Yet scientific communities still desperately lack the concerted effort needed to reduce language barriers faced by non-native English speakers and promote equity in science.

The difficulties faced by non-native English speakers in conducting science, and how they translate to numerous disadvantages for career development, are still poorly understood. Earlier studies have reported the experience and perception of language barriers in speakers of a single non-English language [15] or to certain types of scientific activities, such as paper writing [16], paper publication [17], and research dissemination [18]. Attempts to assess the disadvantages of being non-native English speakers in science are emerging (e.g., [19,20]). Nevertheless, to date, no published study has quantified how multiple aspects of language barriers concurrently affect the career development of speakers of different non-English languages, compared to native English speakers.

This study addresses this knowledge gap by first estimating the amount of effort (e.g., time and financial cost) required by individual researchers in conducting a variety of scientific activities in English. We compare the estimated amount of effort between researchers from countries with different linguistic and economic backgrounds, with the aim to quantify the multiple disadvantages faced by non-native English speakers practising science.

We conducted an online survey of a total of 908 researchers in environmental sciences (particularly ecology, evolutionary biology, conservation biology, and related disciplines) who have published at least one first-authored peer-reviewed paper in English, with one of the following 8 nationalities: Bangladeshi (*n* = 106), Bolivian (100), British (112), Japanese (294), Nepali (82), Nigerian (40), Spanish (108), and Ukrainian (66) (see more details including their demographic information in S1 Table). These nationalities are stratified by the level of each country's English proficiency (based on the English Proficiency Index [21]) and income (based on the World Bank list of economies [22]): Bangladeshi, Nepali (low English proficiency and lower-middle income), Japanese (low English proficiency and high income),

Bolivian, Ukrainian (moderate English proficiency and lower-middle income), Spanish (moderate English proficiency and high income), Nigerian (English as an official language and lower-middle income), and British (English as an official language and high income). This is to distinguish the effect of language barriers from the effect of other types of barriers in science that are often confounded with language barriers, notably economic barriers to conference participations [23,24]. The survey asks participants about the amount of effort needed to conduct 5 categories of scientific activities: paper reading, writing, publication, and dissemination, and participation in conferences (see Materials and methods for more details and S1 Text for the survey itself).

The results unveiled profound disadvantages for non-native English speakers in conducting all scientific activities surveyed. First, non-native English speakers require more time to read an English-language paper—a requisite for obtaining necessary, especially cutting-edge, knowledge in research (Fig 1A and S2 Table). In a comparison among researchers who have published only one English-language paper, non-native English speakers of moderate English proficiency nationalities spend a median of 46.6% (2.5 to 97.5 percentiles: 19.0% to 78.1%) more time, and those of low English proficiency nationalities spend a median of 90.8% (60.6% to 125.4%) more time reading an English-language paper than native English speakers do (Figs 1A and S1). This disadvantage is found even in mid- and late-career researchers, especially those of low English proficiency nationalities (Figs 1A and S1). Importantly, in a comparison of the estimated time needed to read a paper written in their first language, non-native English speakers were shown to need less time than native English speakers (Fig 1B and S3 Table), showing that the above disadvantage arises from the need to read in English, not in their first languages.

Similarly, non-native English speakers need more time to write a paper in English, than their native English speaker peers, at an early career stage (Fig 1C and S4 Table). In a comparison of researchers who have published only one English-language paper, non-native English speakers of moderate English proficiency nationalities spend a median 50.6% (2.5 to 97.5 percentiles: 31.1% to 52.6%) more time, and those of low English proficiency nationalities spend 29.8% (6.6% to 59.3%) more time writing a paper in English than native English speakers do (Figs 1C and S2). This disadvantage is not found in those at a later career stage (S2 Fig). Again, non-native English speakers need less time to write a paper in their first languages than native English speakers do (Fig 1D and S5 Table). This signifies that the need to write in English, not in their first languages, poses a disadvantage to non-native English speakers.

Non-native English speakers also require more effort than native English speakers for the English proofreading of their papers. Apart from late-career researchers of moderate English proficiency nationalities, non-native English speakers ask someone to proofread their English for, on average, 75% or more of their papers, while most native English speakers do this in less than half of their papers (S3 Fig and S6 Table). Non-native English speakers of moderate English proficiency nationalities tend to ask someone to proofread their English as a favour (Fig 1E and S7 Table), while those of a low English proficiency nationality and high income level (i.e., Japanese in our study sample) tend to use a professional English editing service (Fig 1F and S8 Table). Non-native English speakers of low English proficiency nationalities and lower-middle income level neither ask someone to proofread their English as a favour nor use a paid service for most of their papers (Fig 1E and 1F).

Non-native English speakers, especially those of low English proficiency nationalities, are more likely to have their papers rejected by journals due to English writing, compared to native English speakers (Fig 2A and S9 Table). For example, in a comparison of those who have published one English-language paper, 38.1% (31.6% to 44.5%) and 35.9% (30.5% to 41.3%) of the non-native English speakers of moderate and low English proficiency nationalities,

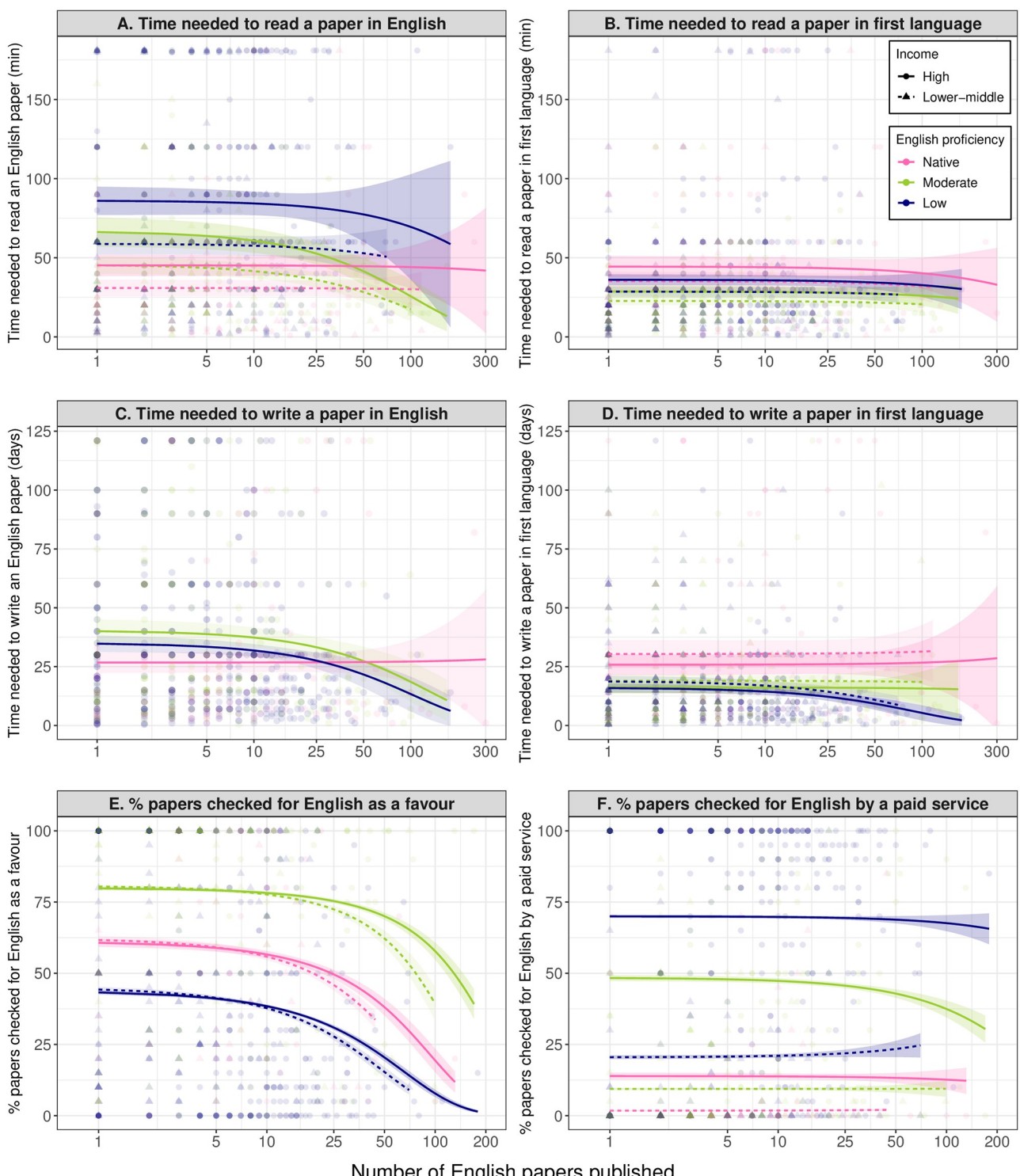

**Fig 1. Language barriers in paper reading and writing.** (**A**) Minutes taken to read and understand the content of the most recent English-language research article each participant read in their field. (**B**) Minutes it would take to fully read and understand the same paper in one's first language. (**C**) Number of days (assuming 7 hours being spent per day) taken to write the first draft of each participant's latest first-authored paper in English. (**D**) Number of days that would be taken to write the first draft of the same paper in their first language. (**E**) Percentage of papers where English writing was checked by someone as a favour. (**F**) Percentage of papers where English writing was checked by a professional service. The regression lines (with 95% confidence intervals as shaded areas) represent the estimated relationship with the number of English-language papers published (shown on the log$_{10}$-transformed axis), based on the results shown in S2–S5 and S7–S8 Tables (income level was not significant in (**C**)). The data underlying this figure are raw data directly from the survey questions, which our ethics approval prevents us from sharing to secure confidentiality of the respondents.

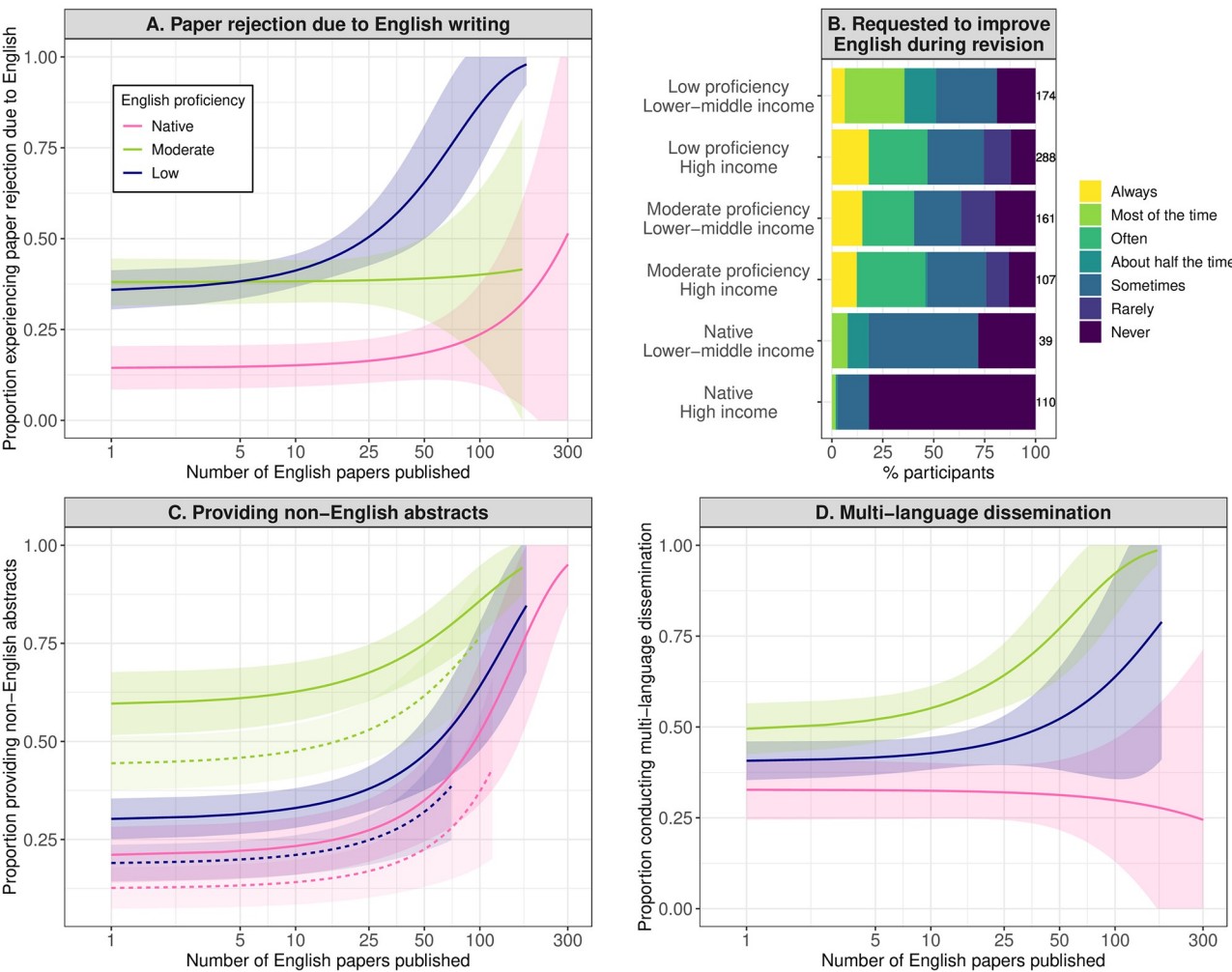

**Fig 2. Language barriers to paper publication and dissemination.** (**A**) Proportion of researchers who have experienced rejection of a first-authored English-language paper due to English writing. (**B**) Frequency of being requested to improve English writing during the revision of first-authored English-language papers. (**C**) Proportion of researchers who have provided non-English-language abstracts of English-language papers. (**D**) Proportion of researchers who have disseminated English-language papers in other languages as well as English. The regression lines (with 95% confidence intervals as shaded areas) in (**A**), (**C**), and (**D**) represent the estimated relationship with the number of English-language papers published (shown on the $\log_{10}$-transformed axis), based on the results shown in S9, S11 and S12 Tables. Income level (solid line: high; dotted line: lower-middle) was only significant and thus shown in (**C**). The data underlying (**A**), (**C**), and (**D**) are raw data directly from the survey questions, which our ethics approval prevents us from sharing to secure confidentiality of the respondents. The data underlying (**B**) can be found in S1 Data.

respectively, have experienced paper rejection due to English writing, while only 14.4% of the native English speakers have, meaning that the frequency of language-related paper rejection is 2.5 to 2.6 times higher for non-native speakers. This result also supports the findings of recent papers that journals are less likely to accept papers by researchers in countries where English is not a primary language [25–27]. Similarly, non-native English speakers are more likely to be requested to improve their English writing during paper revision (Fig 2B and S10 Table). For example, 42.5% and 42.6% of the non-native English speakers of moderate and low English proficiency nationalities, respectively, compared to only 3.4% of the native English speaker population, report that they are often/most of the time/always requested to improve their English writing during paper revision. This equates to a 12.5 times higher frequency of language-related revisions for non-native English speakers.

Non-native English speakers spend more effort disseminating their research in multiple languages than native English speakers do, may it be through the publication of their work in non-English-language journals (S4 Fig), preparation of non-English-language abstracts of English-language papers (Fig 2C and S11 Table), or outreach activities in 2 or more languages (Fig 2D and S12 Table).

Language can also be a major barrier to non-native English speakers attending conferences. Approximately 30% of the early-career (defined as those who have published 5 or fewer English-language papers) non-native English speakers of high income nationalities (i.e., Japanese and Spanish combined) report that they often or always decide not to attend an English-language conference due to language barriers (Fig 3A and S13 Table). Similarly, about half of the early-career non-native English speakers of high income nationalities (Japanese and Spanish combined) often or always avoid oral presentations due to language barriers (Fig 3B and S14 Table).

Even if they decide to give an oral presentation in English, non-native English speakers need much more time to prepare the presentation than native English speakers do; those of moderate and low English proficiency nationalities spend a median 93.7% (2.5 to 97.5 percentiles: 54.7% to 145.2%) and 38.0% (10.8% to 69.6%) more time, respectively, preparing an oral presentation in English than native English speakers do (Fig 4A and S15 Table). This disadvantage does not change with one's career level (S5 Fig) and, yet again, does not apply when preparing a presentation in one's first language. For example, non-native English speakers of low English proficiency nationalities even spend less time preparing a presentation in their first language than native English speakers (Fig 4B and S16 Table). At conferences, non-native English speakers often struggle to explain their work in English. This tendency is particularly noticeable in early-career non-native English speakers of low English proficiency nationalities, with over 65% reporting that they often or always find it difficult to explain their work confidently in English (Fig 4C and S17 Table).

This study illustrates how a series of language barriers to conducting different scientific activities multiply to pose a profound disadvantage to non-native English speakers in the development of their scientific careers (Fig 5). Imagine being a PhD student whose first language is not English. Compared to a fellow student who is a native English speaker, you would need considerably more time, or financial cost, to understand every single English-language paper you read (causing you to spend up to 19.1 more working days per year on this activity. See S1 Fig for the calculation), to write your thesis chapters in English, and to polish the English writing before submitting your manuscripts to journals. You would also struggle with paper publication, as your papers will be rejected more often and be subject to revisions based on the written English. Following the publication of your papers, you would need to make an extra effort for dissemination, as you will be doing this in English as well as your own language(s). You will also find yourself hesitating to attend an international conference, or give an oral presentation, ending up losing opportunities to develop an international network. When you do decide to give an oral presentation, you would again need more time than native English speakers for its preparation, after which you would still be frustrated as you are unable to present your work as effectively in English as you would in your first language. What is more, all of these barriers will continue to get in your way as long as you remain in a research career.

Given all of these disadvantages, all else being equal, the apparent scientific productivity of non-native English speakers would undoubtedly be much lower than that of native English speakers. These disadvantages inevitably lead to a tremendous inequality in the development of scientific careers between native and non-native English speakers and the severe underrepresentation of research from countries where English is not a primary language in English-language publications [28]. Furthermore, at a bigger scale, one clear consequence of this

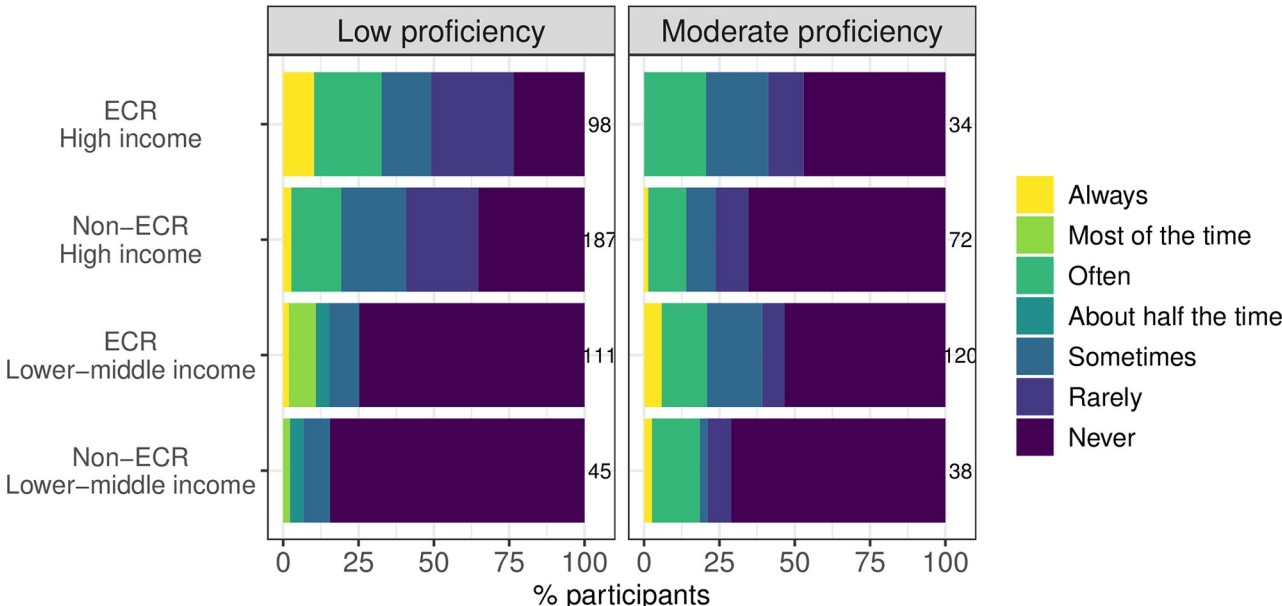

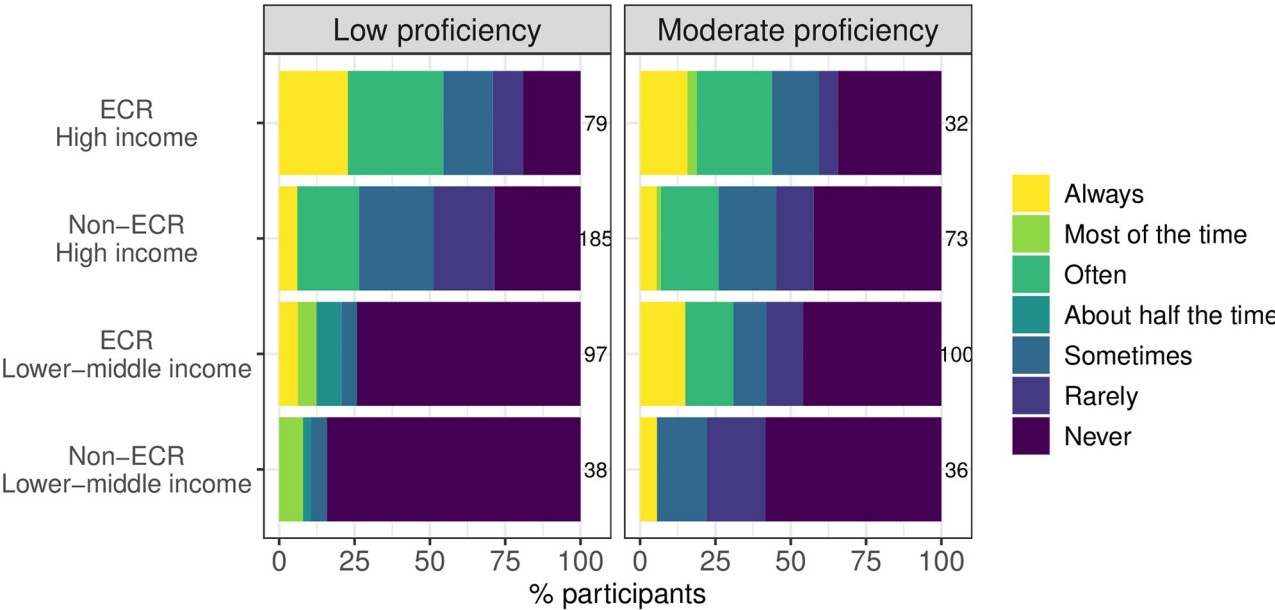

**Fig 3. Language barriers to participation in conferences.** The frequency of (**A**) not attending an English-language conference and (**B**) avoiding oral presentations at an English-language conference due to the lack of confidence in English-language communication. An ECR (early-career researcher) was defined as someone with 5 or fewer English-language papers. The numbers on the right of each bar represent the sample size. The data underlying this figure can be found in S1 Data.

inequality is the loss of opportunity for scientific communities to incorporate many researchers and associated knowledge in the early stages of their careers, partly because their first language happens to be one other than English. This may be reflected in our observation that some disadvantages seemed to disappear in late-career researchers (S1 and S2 Figs) We suspect

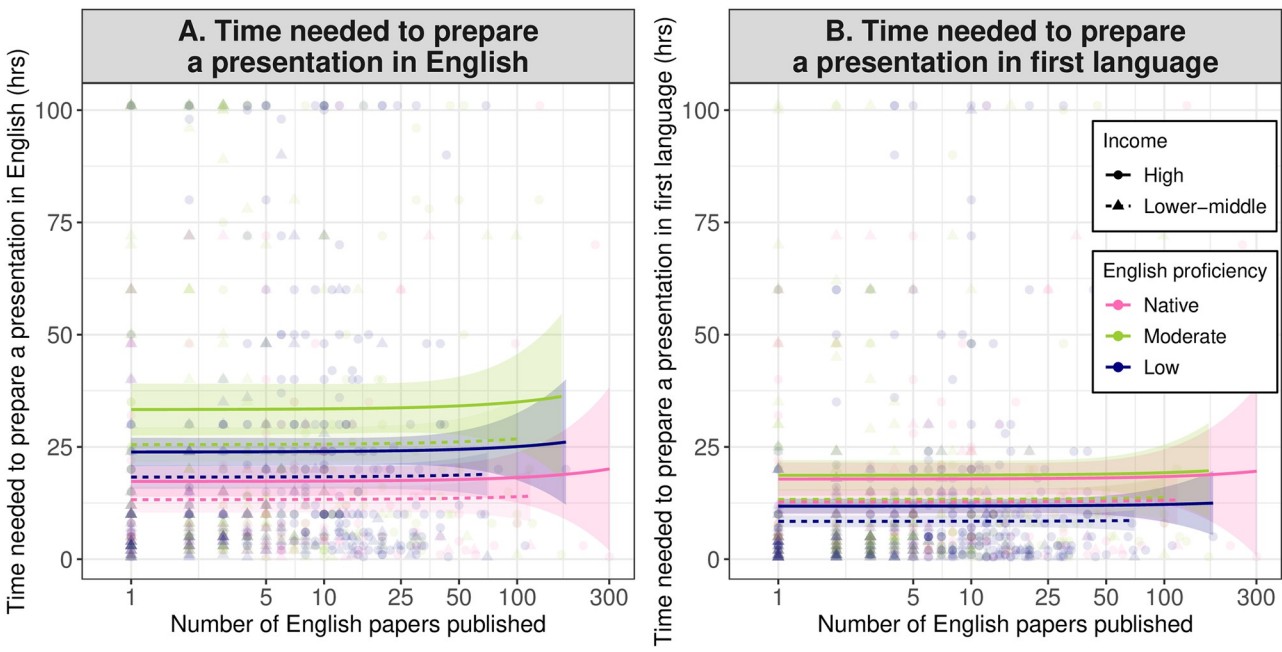

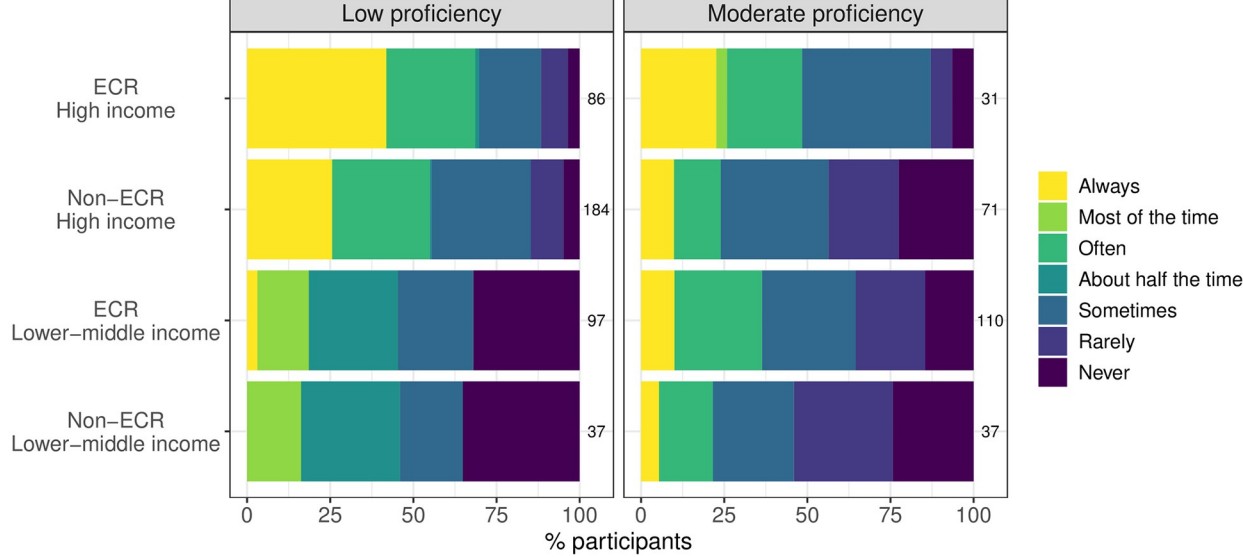

**Fig 4. Language barriers to preparing and conducting presentations in English.** (**A**) Number of hours needed to prepare and practice an oral presentation in English. (**B**) Number of hours that would be needed to prepare and practice the same oral presentation in one's first language. (**C**) Frequency of not being able to explain research confidently during a presentation due to English-language barriers. The regression lines (with 95% confidence intervals as shaded areas) in (**A**) and (**B**) represent the estimated relationship with the number of English-language papers published (shown on the $\log_{10}$-transformed axis), based on the results shown in S15 and S16 Tables. In (**C**), an ECR (early-career researcher) was defined as someone with 5 or fewer English-language papers published so far. The numbers on the right of each bar represent the sample size. The data underlying (**A**) and (**B**) are raw data directly from the survey questions, which our ethics approval prevents us from sharing to secure confidentiality of the respondents. The data underlying (**C**) can be found in S1 Data.

this could be due to survivorship bias; only those non-native English speakers who have managed to conduct science in English as efficiently as native English speakers may have remained in a research career and thus been the dominant group among the experienced researchers who participated in this survey.

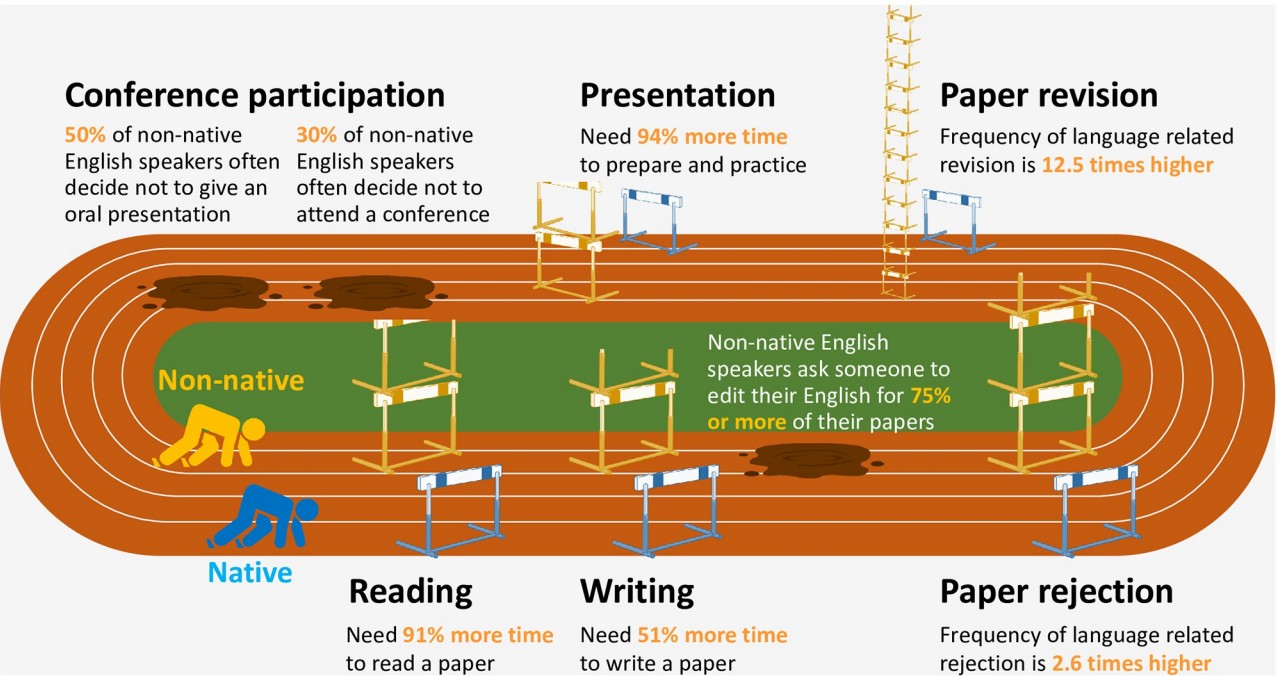

**Fig 5. Estimated disadvantages for non-native English speakers when conducting different scientific activities.** The height of hurdles indicates the relative length of time taken to read an English-language paper (Reading), to write a paper in English (Writing), and to prepare an oral presentation in English (Presentation), and the relative frequency of an English-language paper being rejected (Paper rejection) or requested to revise (Paper revision) due to English writing, for non-native English speakers (Non-native), compared to native English speakers (Native). The values are for non-native English speakers who have published only one English-language paper (higher value from moderate and low English proficiency nationalities), compared to the values for native English speakers. This figure is not intended to suggest that science is a race.

The underuse of professional English editing services by those of lower income nationalities, presumably due to the lack of funding, indicates that disadvantages for non-native English speakers could be amplified by a country's and individual's low income level. Language barriers to some scientific activities, such as reading papers (Fig 1A), preparing oral presentations (Fig 4A), and attending and presenting at conferences (Figs 3A, 3B and 4C), appear to be less severe for those of lower income nationalities. This might again be explained by survivorship bias. Apart from those languages spoken in high-income countries, such as Spanish and Japanese, few non-English languages have an up-to-date lexicon of scientific terms, creating a much higher need for their speakers to receive scientific education in English [29]. In the low-income countries, only those who can afford to receive such English-language education may have been able to become researchers and participate in our survey.

This study is still likely to have underestimated the severity of the disadvantages faced by non-native English speakers. For example, we did not quantify the immense mental stress associated with all the extra time, cost, effort, and lost opportunities caused by language barriers, which could further exacerbate the already high risk of mental health issues in students and early-career researchers [30]. Non-native English speakers could also face the dilemma of adapting to conducting and communicating science in English or maintaining their skills in conducting and communicating science in their first languages [29]. The survey participants are most likely to be those who are currently active in research, and thus the survey has likely excluded those who have dropped out due to language barriers. Other biases in survey participants may also exist (see Limitations in Materials and methods for discussion). Although the survey was designed to isolate the disadvantages associated solely with language barriers, we

cannot dismiss the possibility that the cost we have quantified may incorporate, at least partly, the cost associated with other barriers in science, such as economic, social, identity, and immigration barriers, which many scholars from countries where English is not widely spoken often experience [23,24,31]. While this may be a potential limitation of this study, what this implies in practice is that the disadvantages faced by non-native English speakers could be even bigger and more multifaceted. The level of disadvantages for non-native English speakers could vary among disciplines, presumably depending on, for example, the history of English-based education and the need for international collaboration. Therefore, while we believe that the issue of language barriers to the career development of non-native English speakers is pervasive, the findings of this study may not be quantitatively applicable to all disciplines.

To date, the task of overcoming language barriers has largely been left to non-native English speakers' efforts and their investment in ways of improving their English skills. However, the magnitude of the disadvantage, quantified in this study, seems far beyond the level that can be overcome with individuals' efforts. We urgently need a concerted effort, at institutional and societal levels, to minimise the disadvantages for non-native English speakers. We argue that every sector in science, from supervisors and collaborators to universities, institutions, journals, funders, and conferences, should take immediate action to provide language-related support to non-native English speakers and explicitly take into account those disadvantages when evaluating their scientific outcomes (see Fig 6 for proposed solutions). A key aspect of those solutions is to embrace linguistic diversity in science and encourage the multilingualization of science and its communication, as this can help to improve equity, diversity, and inclusiveness in science [14] and maximise the contribution of science to addressing some of the global challenges [32,33]. Our survey showed the relatively low use of machine translation by researchers in all countries (S6 Fig). However, emerging artificial intelligence (AI) tools, such as ChatGPT (https://chat.openai.com/) and DeepL (https://www.deepl.com/), could help non-native English speakers, especially those in low-income countries, reduce the amount of effort and

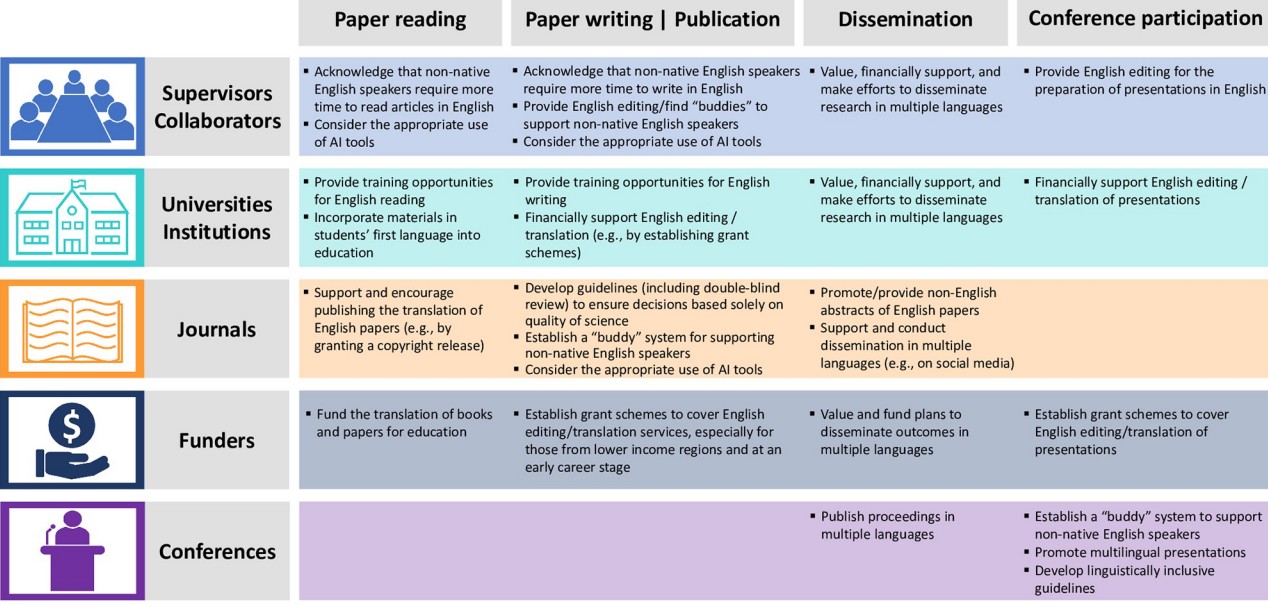

**Fig 6. Examples of potential solutions to reducing disadvantages for non-native English speakers in each type of scientific activities.** AI, artificial intelligence. Also see [35,38,39] for other potential solutions.

the cost needed to do some of the scientific activities, by providing free, or affordable English proofreading/translation [34,35]. Although discussions are still ongoing about the use of generative AI in science [36,37], we believe that journals and universities should consider and allow the appropriate use of AI tools for English proofreading to reduce language barriers and improve equity in science.

The inequality faced by non-native English speakers due to language barriers can be a major reason for the current underrepresentation of non-native English speakers in global scientific activities [40]. One comment from a survey participant caught our eyes:

*If it wasn't for the language barrier, I could have made a much greater contribution to the advance of ecology and biodiversity conservation.* (female participant from Japan in the 40 to 50 age bracket).

Non-native English speakers constitute 95% of the world population [41]. Imagine how many non-native English speakers around the world and over time have been frustrated, just like this participant, because they are unable to contribute to the advance of science to the best of their abilities. Think how many potential contributors scientific communities have failed to bring onboard due to language barriers. Given the multitude of pressing challenges facing humanity and this planet, surely, we cannot afford to miss contributions from such a promising, much needed, yet currently untapped source of researchers.

## Materials and methods

### Ethics statement

The survey was conducted between June and October 2021 in accordance with the University of Queensland's Institutional Human Research Ethics Approval (committee: Science Low and Negligible Risk Committee, approval number: 2021/HE000566). All participants were at least 18 years old and provided written consent indicating their agreement to participate in the survey. The Participant Information Sheet clarified the voluntary nature of participation, the aims of the research, how the data would be used, and that all data would be confidential.

The aim of the survey was to (i) quantify the amount of effort required by individual researchers to conduct 5 types of scientific activities in English and their first language: paper reading, writing, publication, and dissemination, and participation in conferences; and (ii) compare the estimated amount of effort between researchers with different linguistic and economic backgrounds.

### Target participants

For the comparison between researchers with different linguistic and economic backgrounds, we selected 8 nationalities: Bangladeshi, Bolivian, British, Japanese, Nepali, Nigerian, Spanish, and Ukrainian. These nationalities were stratified by the levels of each country's English proficiency (based on the English Proficiency Index [21]) and income (based on the World Bank list of economies [22]): Bangladeshi, Nepali (low English proficiency and lower-middle income), Japanese (low English proficiency and high income), Bolivian, Ukrainian (moderate English proficiency and lower-middle income), Spanish (moderate English proficiency and high income), Nigerian (English as an official language and lower-middle income), and British (English as an official language and high income). We focused on English proficiency and income level based on our hypothesis that the amount of effort needed to conduct scientific activities in English would be higher in non-native English speakers from countries with lower English proficiency and income level.

Note that the level of countries' English proficiency does not necessarily reflect the level of each participant's English proficiency. However, the level of countries' English proficiency was

significantly related to 2 of the 3 other measures of participants' experience in English communication: the percentage of time spent speaking English in a day and the number of years spent living in countries where English is the first language (S7–S9 Figs). This supports the use of countries' English proficiency as a crude measure of participants' English proficiency.

Countries' income levels do not necessarily reflect each participant's socioeconomic level either. This study is thus not able to assess the effect of individuals' socioeconomic backgrounds.

The survey was targeted at anyone at any career level and of any profession who has the selected countries' nationality and has published at least one first-authored peer-reviewed English-language paper in ecology, evolutionary biology, conservation biology, or related disciplines.

### Questionnaire survey

The survey (provided in S1 Text) consists of 6 sections. The first section (Q1.1 to Q1.2) is about participants' first language (defining participants' first language is admittedly difficult in some countries, such as Nigeria, but we used the following definition: "the language(s) you learnt to speak at home as a child") and nationality; this information was used to filter for eligible participants. The second section (Q2.1 to Q2.7) comprises questions on background information including measures of English proficiency; these were used to account for factors that may affect the answers to the other questions in the survey during analysis and to justify the use of countries' English proficiency in the analysis. The third section (Q3.1 to Q3.7) includes questions on participants' experience of language barriers when writing papers in English. The fourth section (Q4.1 to Q4.5) asks about participants' experience of language barriers in paper publication and dissemination. The fifth section (Q5.1 to Q5.3) is about the consequences of language barriers to paper reading in English, and the sixth section (Q6.1 to Q6.6) asks how language barriers might have affected participants' experiences around the attendance of scientific conferences. The survey also allowed participants to give comments on the survey as well as general feedback on the project.

To allow participants to estimate the length of time required to do each scientific activity as accurately as possible, we asked participants to provide data on actual experiences, i.e., how long it took them to write the latest paper that they wrote (Q3.3), read the latest paper that they read (Q5.1), and prepare the latest oral presentation that they gave (Q6.4) in English. We also asked non-native English speakers to estimate the length of time that would be required to write the same paper (Q3.4), read the same paper (Q5.2), and prepare the same presentation (Q6.5) but in their first language. See Limitations for a discussion on the potential consequences of this approach for deriving conclusions. When asking frequency, we used a 5-point Likert scale: Always, Often, Sometimes, Rarely, or Never. Our questions were also designed to ask participants about their experiences that were solely due to language barriers, and not other, often confounding barriers (e.g., by stating the part in bold: "*6.2 How often have you decided not to attend an English-language conference (either for presenting your research or just for participating)* ***because you were not confident enough to communicate in English?***". Also see other questions in S1 Text).

To maximise the response rate, the survey was translated into the relevant languages for each nationality (Bangla for Bangladeshi translated by SC, Japanese for Japanese by TA, Nepali for Nepali by KP, Spanish for Bolivian and Spanish by VB-E, and Ukrainian for Ukrainian by MG) and implemented as a separate online survey for each nationality on Qualtrics. We created a unique link and QR code for each country, which was used for distribution described below.

## Survey distribution

We first identified coordinators (hereafter referred to as country coordinators) for each of the 8 selected countries, who (i) is a native speaker of the official language of the country and (ii) has a good network among researchers in the relevant disciplines in the country. All country coordinators were involved in this study as coauthors (TA for Japan, IB for Nigeria, SC for Bangladesh, MG for Ukraine, JDG-T for Spain, FM-C for Bolivia, KP for Nepal, and RLW for the United Kingdom). Country coordinators aimed to collect responses to the survey from at least 100 participants in each country. We tried to distribute the survey in as unbiased a way as possible. To achieve this, we adopted, in principle, one or all of the following 4 methods of survey distribution within each country, based on discussions with each country's coordinator on which method(s) might be the best for that country:

- Distribute the survey through major mailing list(s) for researchers in relevant disciplines.

- Ask academic societies of relevant disciplines to distribute the survey to their members.

- Identify up to 10 universities and institutions with relevant departments, schools, or divisions within the country and ask them to distribute the survey to their affiliated researchers.

- Identify researchers who have published an English-language paper in a relevant discipline and are affiliated to an institution in the country on literature search systems and directly send the survey to them via email.

We avoided using our personal networks (including personal social media accounts) to disseminate the survey as much as possible, in order to reduce potential biases in participant recruitment (but see exceptions for Bangladesh below). The detailed method of survey distribution in each country is described below (all dates refer to 2021).

**Bangladesh.** In Bangladesh, we could not find any relevant mailing lists. Academic societies exist but early-career researchers do not necessarily belong to those societies, and we thus decided not to distribute the survey through academic societies either. Instead, the survey was distributed by directly contacting 7 universities and a total of 232 individual researchers identified on Google Scholar and Facebook.

22nd and 27th June: Shared the survey on the country coordinator's personal Facebook account.

14th to 18th July: Contacted representatives at 4 major universities (University of Dhaka, Jahangirnagar University, Pabna University of Science and Technology, and Noakhali Science and Technology University) and asked them to share the survey within their relevant departments.

25th July: Recontacted representatives at 3 universities (University of Dhaka, Jagannath University, and Noakhali Science and Technology University) and asked them to share the survey within their relevant departments. Also emailed a professor at the University of Dhaka to share the survey with colleagues, who also shared it with many other academics in the country.

31st July: Recontacted a representative at the University of Dhaka and newly contacted representatives at 3 more universities (Sher-e-Bangla Agricultural University, Bangladesh Agricultural University, and Chittagang University) and asked them to share the survey within their relevant departments.

8th August: Reshared the survey on the country coordinator's personal Facebook and Twitter account.

12th September: Directly emailed the survey to the top 100 Bangladeshi researchers identified on Google Scholar (searched with (conservation OR ecology OR evolution) AND Bangladesh).

22nd September to 15th October: Contacted 120 researchers in relevant disciplines identified on Facebook.

28th October: Shared the survey on the country coordinator's personal Facebook and LinkedIn accounts and also contacted 12 researchers while sending a reminder to those who were already contacted.

**Bolivia.** In Bolivia, the survey was distributed through a major mailing list and by contacting 4 societies, 5 universities, 4 museums/herbaria, and a total of 72 individual researchers identified on the Web of Science.

29th June: Shared the survey on a major mailing list for biologists and ecologists in Bolivia. Reminders sent once within June and another in July. The survey was also sent to the Organization of Women in Science Bolivia, the Bolivian Association of Ornithologists, the Bolivian Association of Mammalogists, and the Bolivian Society of Entomologists, for sharing on their mailing lists.

1st July: Contacted the Heads of the Departments of Biology, Zoology, Botany, and Ecology in all 5 universities that have a science department in Bolivia (Universidad Mayor de San Andrés, Universidad Amazónica de Pando, Universidad Mayor Gabriel Rene Moreno, Universidad Mayor de San Simón, and Universidad San Francisco Xavier de Chuquisaca) and the 4 major museums/herbaria in Bolivia (Colección Boliviana de Fauna, Herbario Nacional de Bolivia, Museo de Historia Natural Noel Kempff Mercado, and Museo Nacional Martin Cardenas) and asked them to share the survey within their departments. Sent reminders to them on 26th July.

16th September: Searches were conducted on Web of Science (using all databases) with: ALL = ((conservation OR ecolog* OR evolution*) AND (Bolivia)). A total of 3,715 studies were returned from the search, from which 72 first authors who seemed to be Bolivians were identified. The survey was directly shared with the 72 authors via email. For those authors who were not accessible through the email addresses on the papers, the country coordinator looked for their new contact addresses (on ORCID and some other platforms) and, if found, used the new addresses to contact them.

**Japan.** In Japan, the survey was shared via 2 major mailing lists.

9th June: Shared the survey on the 2 major mailing lists for ecologists (*jeconet*, with 3,500 users as of 2014) and evolutionary biologists (*evolve*, with 2,500 users as of 2016) in Japan.

23rd June: Sent a follow-up email to the same 2 mailing lists.

**Nepal.** In Nepal, the survey was shared with 5 societies and 5 universities.

2nd July: Asked the Nepal Environment Society, the Environmental Graduates in Himalaya, the Society for Conservation Biology Nepal Chapter, the Botanical Society of Nepal, and the Zoological Society of Nepal (altogether these societies have more than 600 members) to share the survey on their mailing lists.

27th July: Sent reminders to those who were contacted above.

5th September: Contacted the Heads of Departments of 5 universities that have programmes in biodiversity conservation and natural sciences (Kathmandu University, Tribhuvan

University, Pokhara University, Mid-western University and Agriculture, and Forestry University) over the phone and asked them to share the survey within their departments.

20th September: Sent reminders to those universities.

**Nigeria.** In Nigeria, the survey was distributed by contacting 3 relevant societies, 3 institutes with relevant departments, 5 universities (from 5 of the 6 geopolitical zones in Nigeria), and a total of 54 individual researchers identified on Google Scholar.

21st June: Shared the survey with the Nigerian Tropical Biology Association alumni group, scientists at the National Center for Genetic Resources and Biotechnology, and researchers at the Department of Zoology, University of Lagos.

22nd and 23rd June: Shared the survey with scientists at the Sheda Science and Technology Complex.

6th July: Contacted the assistant secretary of the Zoological Society of Nigeria, who shared the survey with all of the society's members (approximately 400 people).

8th July: Shared the survey with 36 faculties across the Departments of Botany, Forest Resources Management, Wildlife and Ecotourism, Chemistry, Geography, and Geology at the University of Ibadan.

10th July: Shared the survey on WhatsApp among all scientists of the Cocoa Research Institute of Nigeria, a federal government institution with over 200 research staff.

14th July: Sent reminders to the Nigerian Tropical Biology Association alumni group, scientists at the National Center for Genetic Resources and Biotechnology, and researchers at the Department of Zoology, University of Lagos.

12th September: Shared the survey with 60 faculty members of the Adekule Ajasin University and one at the Abubakar Tafawa Balewa University.

14th October: Shared the survey with 63 faculty members of Ahmadu Bello University.

18th October: Shared the survey with 173 members of the Society for Conservation Biology Nigerian Chapter, and 54 authors identified through searches on Google Scholar using: "(conservation OR ecology OR evolution) AND Nigeria".

**Spain.** In Spain, the survey was shared with 5 societies, 19 universities, and a museum. We chose 1 to 4 universities with a strong biology department from each of the 9, out of the 17, autonomous communities of Spain, so that the selected universities are geographically scattered.

21st June: Asked the Limnological Society, the Society of Terrestrial Ecologists, the Society for Evolutionary Biology, the Society for Biochemistry and Molecular Biology, and the Society for Cellular Biology to share the survey with their members via their channels.

5th July: Sent the first reminder to the 5 societies above.

30th August: Sent a second reminder to the 5 societies. Asked the biology/science departments of 9 universities across the country to share the survey within their departments: Universidad de Barcelona, Universidad Autónoma de Barcelona, Universidad de Girona, Universidad Complutense de Madrid, Universidad de Sevilla, Universidad de Valencia, Universidad de Cádiz, Universidad de Murcia, and Universidad del País Vasco.

13th September: Sent a third reminder to the first 5 societies, and the first reminder to the 9 additional universities.

4th October: Sent a fourth reminder to the 5 societies, and a second reminder to the 9 universities. Asked 10 additional universities and a museum to share the survey within their networks: Universidad del Rey Juan Carlos, Universidad Autónoma de Madrid, Universidad de Salamanca, Universidad de Huelva, Universidad de Málaga, Universidad de Burgos, Universidad de León, Universidad de Castilla y La Mancha, Universidad de Alicante, Universidad de Zaragoza, and Madrid's Museum of Natural Sciences.

18th October: Sent reminders to the 5 societies, 19 universities, and the museum.

25th October: Sent reminders to the 5 societies, 19 universities, and the museum.

**Ukraine.** In Ukraine, the survey was shared through 10 universities, 3 institutes, 3 Facebook groups, and a total of 139 individual researchers identified on the Web of Science, conference abstracts, and Ukrainian journals.

29th June: Shared the survey among employees of the State Museum of Natural History (Lviv); also posted on the Facebook group Flora of Ukraine by the museum administrator. Asked the Institute of Ecology of the Carpathians, NASU (Lviv) to share the survey within their network.

22nd July: Asked the I.I. Schmalhausen Institute of Zoology of the National Academy of Sciences of Ukraine (NASU) (Kyiv) to share the survey within their network.

13th September: Shared the survey with all researchers at the Institute of Marine Biology, NASU (Odesa), and 139 researchers identified on the Web of Science (using keywords: All = ((conservation OR ecolog* OR evolution*) AND (Ukraine))) and by searching for conference abstracts on Google (using keywords: "еволюційна біологія конференція", "охорона природи конференція", or "екологія конференція").

14th September: Asked biology/ecology departments of 10 universities (Khmelnytsky National University, Petro Mohyla Black Sea National University, Sumy State University, National University of Water and Environmental Engineering, National University of Life and Environmental Sciences of Ukraine, Poltava National Agricultural University, Ukrainian National Forestry University, Ivano-Frankivsk National Technical University of Oil and Gas, Chernivtsi National University, and National Museum of Chernivtsi National University) to share the survey within their network.

27th September: Sent reminders to all individual researchers who were contacted on 13th September.

11th October: Sent reminders to all individual researchers who were previously contacted.

11th October: Shared the survey in the Facebook group Ukrainian Botanical Group.

13th October: Shared the survey in the Facebook group Ukrainian Scientists Worldwide.

**United kingdom.** In the UK, the survey was disseminated through 3 societies/professional bodies, 1 research institute, and 20 universities.

- British Ecological Society (BES)

  10th June: Asked to disseminate the survey via their channels.

  25th August: Sent a reminder.

  The BES journals' twitter accounts tweeted about the survey:

7th July and 7th September @MethodsEcolEvol (26.3k followers).

13th July and 13th September @FunEcology (21.6k followers).

14th July and 7th September @jecology (30.7k followers).

9th July and 7th September @JAppliedEcology (31.4k followers).

7th July and 7th September @AER_ESE_BES (2.1k followers).

7th July and 7th September @AnimalEcology (22.7k followers).

7th July and 15th September @PaN_BES (4.6k followers)

- Royal Society of Biology (RSB)

10th June: Asked to disseminate the survey via their channels.

25th June: The survey was shared in their Science Policy Newsletter, which goes out to roughly 26,000 people, most in the UK.

25th August: Sent a reminder.

10th September: The survey was shared again in their Science Policy Newsletter.

- Chartered Institute of Ecology and Environmental Management (CIEEM)

10th June: Asked to disseminate the survey via their channels.

25th August: Sent a reminder

- Centre for Ecology and Hydrology (CEH)

10th June: Asked to disseminate the survey via their channels.

1st September: CEH tweeted about the survey @UK_CEH (39.6k followers)

13th September: CEH tweeted about the survey @UK_CEH

- Universities

1st September: Selected and emailed 10 universities to reach out and request to disseminate the survey internally. Using the 2022 "The Complete University Guide" rankings for Biological Sciences (which includes, but is not limited to, Biological Sciences, Biology, Ecology, Marine Biology, Cell Biology, Microbiology, Plant Sciences, Zoology, Genetics, Biochemistry, Applied Biology, Evolution), every 10th institution within the top 100 universities was selected:

#1 University of Cambridge, School of the Biological Sciences.

#10 University of Glasgow, School of Life Sciences.

#20 University of Leeds, Faculty of Biological Sciences.

#29 University of Nottingham, School of Life Sciences (#30 University of Sunderland was not selected as not appropriate).

#39 University of Kent, Durrell Institute of Conservation and Ecology (#40 Glasgow Caledonian University was not selected as not appropriate).

#49 University of Plymouth, School of Biological and Marine Sciences (#50 Keele University was not selected as not appropriate).

#60 University of Lincoln, School of Life Sciences.

#70 University of Northampton.

#80 Liverpool John Moores University, School of Biological and Environmental Sciences.

#90 University of Derby, School of Built and Natural Environment.

13th September: Sent a reminder to all university departments.

5th October: Sent a reminder to all university departments.

5th October: Reached out to a further 10 universities as follows:

#2 = University of Oxford.

#11 = University of Bristol.

#21 = University of Bath.

#31 = Swansea University.

#41 = Edinburgh Napier University.

#51 = University of Essex.

#61 = Aberystwyth University.

#72 = Bangor University (#71 University of Westminster was not selected as not appropriate).

#81 = University of Brighton.

#91 = University of Suffolk.

## Limitations

The limitations of our survey include (i) relatively small sample size; (ii) potential bias in participant recruitment; and (iii) difficulties in estimating the length of time taken to conduct scientific activities in different languages.

Despite the considerable effort we put in in distributing the survey at 71 universities, 12 institutes, and 23 societies, on 3 mailing lists, and with 497 individual researchers across 8 countries, the sample size of this study (908, ranging from 67 to 292 per language) is not necessarily large. This may have caused the lack of power in our analyses, which could explain the non-significant effect of income level in some analyses.

Although we tried to recruit survey participants in as unbiased a way as possible (see Survey distribution), we acknowledge that the recruited participants are likely to represent non-random samples of the entire eligible population. For example, survey participants are most likely to be active researchers, and thus the survey likely excludes those who have already left their research careers due to language barriers. Our survey also excluded those who have never published a first-authored English-language paper. This could lead to an underestimation of the actual severity of the language barriers experienced by the entire population of non-native English speakers. We also recorded 5 potential covariates that can affect the amount of effort required to conduct scientific activities in English: age, gender, discipline, the number of years in research, and the number of English-language publications. Age, gender, discipline, and the number of years in research were all correlated with the number of English-language publications (see Analyses for more detail). Therefore, we used the

number of English-language publications as a covariate in all analyses, to account for the effect of these covariates.

It is admittedly difficult for participants to estimate the exact length of time taken, or would take, to write a paper, read a paper, or prepare an oral presentation in English and in their first languages. To allow participants to provide as accurate an estimate as possible, we asked them the actual time taken to, for example, write the most recent paper that they wrote in English, rather than the time that they *think* is required to write an imaginary paper, as it is normally easier and more accurate to report the most recent experience (recall bias; see, e.g., [42]). There is no reason to believe that non-native English speakers consistently overestimate the actual length of time they have spent on scientific activities. We rather expect that the difficulty in estimating the length of time taken to conduct scientific activities can affect precision, as is reflected in large variation within each group of the English proficiency–economic level combinations. As we asked the participants to answer based on actual experiences, the reported length of time taken to, for example, write a paper would also have depended on the varying length of the paper. Nevertheless, again, there is no reason to believe that papers written by non-native English speakers are consistently longer than those written by native English speakers. We thus do not believe that these issues affect the main conclusion of this study. That said, the reported length of time it would take to conduct scientific activities in their first language is not based on the participants' actual experience and thus needs to be interpreted with care.

## Analyses

In the analyses, we only used data on participants whose nationalities were one of the 8 target nationalities and whose first language was one of the 6 target languages. In all the analyses, we aimed to test whether the amount of effort required for scientific activities, or the frequency of facing language barriers in science, differs for participants depending on their native country's level of English proficiency and economy, while accounting for the effect of covariates.

As covariates, we considered the following 5 variables: age, gender, discipline, the number of years in research, and the number of English-language publications. We first tested correlations between the 5 covariates. Age and the number of years in research were both highly correlated with the number of English-language publications (Spearman's rank correlation coefficient = 0.58 for age and 0.64 for the number of years in research). There was also a highly significant relationship between gender and the number of English-language publications (Kruskal–Wallis chi-squared = 68.37, $p < 1.42 \times 10^{-15}$) and between disciplines and the number of English-language publications (Kruskal–Wallis chi-squared = 29.45, $p < 6.35 \times 10^{-6}$). Thus, we decided to only use the number of English-language publications as a covariate in the following analyses.

We used 3 types of models depending on the type of the response variables:
Generalised linear models with a negative binomial distribution for

- The number of minutes taken to read and understand the last English-language original article each participant read in their field.

- The number of minutes it would take to read and understand the same paper but in their first language.

- The number of days taken to write the first draft of each participant's latest first-authored paper in English.

- The number of days it would have taken to write the first draft of each participant's latest first-authored paper in their first language.

- The number of hours taken to prepare and practice an oral presentation in English.

- The number of hours it would take to prepare and practice the same oral presentation in their first language.

  Generalised linear models with a binomial distribution for

- The percentage of papers where English writing was checked either by someone as a favour or by a paid service.

- The percentage of papers where English writing was checked by someone as a favour.

- The percentage of papers where English writing was checked by a paid service.

- The experience of a first-authored English-language paper being rejected due to English writing.

- The experience of providing a non-English-language abstract of English-language papers.

- The experience of conducting the dissemination of English-language papers in other language(s) as well as English.

  Cumulative link models for

- The frequency of being requested to improve English writing in the revision of first-authored English-language papers.

- The frequency of not attending an English-language conference due to the lack of confidence in English-language communication.

- The frequency of avoiding giving oral presentations at an English-language conference due to the lack of confidence in English-language communication.

- The frequency of not being able to explain one's own research confidently during a presentation due to English-language barriers.

In all models, we used 3 explanatory variables: a country's English language proficiency (English native as the reference category, moderate (the reference category in analyses not including English natives), and low), a country's income level (high as the reference category, and lower-middle), and the number of English-language publications, as well as 2 interactions: English language proficiency and the number of English-language publications, and income level and the number of English-language publications. We first tested whether the 2 interactions were significant using the likelihood ratio test and excluded any non-significant interactions. If any interaction was excluded, we again tested whether the explanatory variables that were involved in the interaction(s) were significant using the likelihood ratio test and excluded any non-significant variables to determine the final model. We interpreted the results derived from the final models. In a few analyses (shown in S3, S15 and S16 Tables), however, even non-significant variables were retained in the final models to enable comparisons with results from other associated analyses.

All analyses were conducted in R version 4.1.2 [43]. We also used the following R packages: tidyverse [44], MASS [45], lmtest [46], janitor [47], corrplot [48], ordinal [49], and gridExtra [50].

## Supporting information

**S1 Table. Survey participants by nationality and first language.** The gender composition of the participants was 339 female, 556 male, and 13 participants in other categories, with the median age of 39 (range: 18–77) years old and median 13 (range: 1–55) years of experience in research.
(DOCX)

**S2 Table. Results of a generalised linear model (with a negative binomial distribution) of factors explaining variations in the number of minutes taken to read and understand the entire content of the last English-language original article each participant read in their field.** The reference category for English proficiency and Income level was English native and High income, respectively.
(DOCX)

**S3 Table. Results of a generalised linear model (with a negative binomial distribution) of factors explaining variations in the number of minutes it would take to read and under-stand the same paper in full, but in their first language.** The reference category for English proficiency and Income level was English native and High income, respectively. The number of English papers published was not significant in the likelihood ratio test but was retained in the final model for a comparison with the result shown in S2 Table.
(DOCX)

**S4 Table. Result of a generalised linear model (with a negative binomial distribution) of factors explaining variations in the number of days taken to write the first draft of each participant's latest first-authored paper in English.** The reference category for English profi-ciency and Income level was English native and High income, respectively.
(DOCX)

**S5 Table. Result of a generalised linear model (with a negative binomial distribution) of factors explaining variations in the number of days it would take to write the first draft of each participant's latest first-authored paper in their first language.** The reference category for English proficiency and Income level was English native and High income, respectively.
(DOCX)

**S6 Table. Result of a generalised linear model (with a binomial distribution) of factors explaining variations in the percentage of papers where English writing was checked either by someone as a favour or by a paid service.** The reference category for English proficiency and Income level was English native and High income, respectively.
(DOCX)

**S7 Table. Result of a generalised linear model (with a binomial distribution) of factors explaining variations in the percentage of papers where English writing was checked by someone as a favour.** The reference category for English proficiency and Income level was English native and High income, respectively.
(DOCX)

**S8 Table. Result of a generalised linear model (with a binomial distribution) of factors explaining variations in the percentage of papers where English writing was checked by a paid service.** The reference category for English proficiency and Income level was English native and High income, respectively.
(DOCX)

**S9 Table. Result of a generalised linear model (with a binomial distribution) of factors explaining the experience of having a first-authored English-language paper rejected due to English writing.** The reference category for English proficiency and Income level was English native and High income, respectively.
(DOCX)

**S10 Table. Result of a cumulative link model of factors explaining the frequency of being requested to improve English writing in the revision of first-authored English-language papers.** The reference category for English proficiency and Income level was English native and High income, respectively.
(DOCX)

**S11 Table. Result of a generalised linear model (with a binomial distribution) of factors explaining the experience of providing a non-English-language abstract of English-language papers.** The reference category for English proficiency and Income level was English native and High income, respectively.
(DOCX)

**S12 Table. Result of a generalised linear model (with a binomial distribution) of factors explaining the experience of disseminating English-language papers in other language(s) in addition to English.** The reference category for English proficiency and Income level was English native and High income, respectively.
(DOCX)

**S13 Table. Result of a cumulative link model of factors explaining the frequency of not attending an English-language conference due to a lack of confidence in English communication.** The reference category for English proficiency and Income level was Low English proficiency and High income, respectively.
(DOCX)

**S14 Table. Result of a cumulative link model of factors explaining the frequency of avoiding oral presentations at an English-language conference due to a lack of confidence in English communication.** The reference category for English proficiency and Income level was Low English proficiency and High income, respectively.
(DOCX)

**S15 Table. Results of a generalised linear model (with a negative binomial distribution) of factors explaining variations in the number of hours taken to prepare and practice an oral presentation in English.** The reference category for English proficiency and Income level was English native and High income, respectively. The number of English papers published was not significant in the likelihood ratio test but was retained in the final model for a comparison with other results.
(DOCX)

**S16 Table. Results of a generalised linear model (with a negative binomial distribution) of factors explaining variations in the number of hours that would be taken to prepare and practice the same oral presentation in the first language.** The reference category for English proficiency and Income level was English native and High income, respectively. The number of English papers published was not significant in the likelihood ratio test but was retained in the final model for a comparison with other results.
(DOCX)

**S17 Table. Result of a cumulative link model of factors explaining the frequency of not being able to explain research confidently during a presentation due to English barriers.**

The reference category for English proficiency and Income level was Low English proficiency and High income, respectively.
(DOCX)

**S1 Fig. The number of extra minutes (and its 95% confidence intervals as shaded areas) estimated to take researchers of moderate (green) and low (navy) English proficiency nationalities to read and understand the entire content of the last English-language original article they read in their field, compared to native English speakers, in relation to the number of English-language papers published.** The estimations are based on the results of the regression shown in S2 Table. The solid vertical lines (and 95% confidence intervals as broken vertical lines) indicate the number of English-language papers published, as a measure of career level, where non-native English speakers do not take longer to read an English-language paper than native English speakers. Non-native English speakers who have published only one English-language paper were estimated to require, on average, 40.18 (low English proficiency nationalities) and 21.31 (moderate English proficiency nationalities) more minutes to read an English-language article, compared to their native English-speaking counterparts. If they were to read 200 articles per year (average number of article readings per year for US faculty [51]), this equates to 19.1 (low English proficiency nationalities) and 10.1 (moderate English proficiency nationalities) more working days per year, assuming a 7-hour working day. The data underlying this figure can be found in S1 Data.
(PDF)

**S2 Fig. The number of extra days (and its 95% confidence intervals as shaded areas) estimated to take researchers of moderate (green) and low (navy) English proficiency nationalities to write the first draft of their latest first-authored paper in English, compared to native English speakers, in relation to the number of English-language papers published.** The estimations are based on the results of the regression shown in S4 Table. The solid vertical lines (and 95% confidence intervals as broken vertical lines) indicate the number of English-language papers published, as a measure of career level, where non-native English speakers do not take longer to write an English-language paper than native English speakers. The data underlying this figure can be found in S1 Data.
(PDF)

**S3 Fig. The proportion of researchers who have their English writing checked either by someone as a favour or by a professional service.** The regression lines (with 95% confidence intervals as shaded areas) represent the estimated relationship with the number of English-language papers published, based on the results shown in S6 Table. The data underlying this figure are raw data directly from the survey questions, which our ethics approval prevents us from sharing to secure confidentiality of the respondents.
(PDF)

**S4 Fig. Reasons for non-native English speakers to submit their papers to non-English-language journals by nationality.** Participants were allowed to choose multiple reasons, and the x-axis indicates the percentage of participants who selected each reason. The data underlying this figure can be found in S1 Data.
(PDF)

**S5 Fig. The number of extra hours (and its 95% confidence intervals as shaded areas) estimated to take researchers of moderate (green) and low (navy) English proficiency nationalities to prepare and practice an oral presentation in English, compared to native English speakers, in relation to the number of English-language papers published.** The estimations

are based on the results of the regression shown in S15 Table. The data underlying this figure can be found in S1 Data.
(PDF)

**S6 Fig. The frequency of machine translation usage when reading English-language papers by nationality.** The data underlying this figure can be found in S1 Data.
(PDF)

**S7 Fig. The percentage of time spent speaking English, per day, in daily life.** Researchers of moderate English proficiency nationalities speak English in daily life significantly more than those with low English proficiency (generalised linear model with a binomial distribution: Coefficient = 0.35, Standard Error = 0.022, z = 16.40, $p < 2.0 \times 10^{-16}$). The data underlying this figure can be found in S1 Data.
(PDF)

**S8 Fig. The number of years learning English as a foreign language.** Researchers of moderate English proficiency nationalities have been spending a significantly fewer number of years learning English than those of low English proficiency nationalities (generalised linear model with a negative binomial distribution: Coefficient = −0.22, Standard Error = 0.044, z = −4.96, $p = 7.23 \times 10^{-7}$). The data underlying this figure can be found in S1 Data.
(PDF)

**S9 Fig. The number of years lived/living in countries where English is the first language.** Researchers of moderate English proficiency nationalities have lived in a country where English is the first language significantly longer than those of low English proficiency nationalities (generalised linear model with a negative binomial distribution: Coefficient = 0.47, Standard Error = 0.17, z = 2.69, $p = 0.0072$). The data underlying this figure can be found in S1 Data.
(PDF)

**S1 Data. The data underlying Figs 2B, 3A, 3B and 4C, S1, S2, S4, S5, S6, S7, S8 and S9.**
(XLSX)

**S1 Text. Questionnaire survey on consequences of language barriers for non-native English speakers for developing career in science.**
(DOCX)

**S2 Text. Alternative language abstract and Figs 5 and 6 in Japanese.**
(DOCX)

**S3 Text. Alternative language abstract and Figs 5 and 6 in Nepali.**
(DOCX)

**S4 Text. Alternative language abstract and Figs 5 and 6 in Portuguese.**
(DOCX)

**S5 Text. Alternative language abstract and main text in Spanish.**
(DOCX)

**S6 Text. Alternative language abstract and Figs 5 and 6 in Ukrainian.**
(DOCX)

## Acknowledgments

We thank all participants in our survey and M. Amano for English proofreading.

## Author Contributions

**Conceptualization:** Tatsuya Amano, Valeria Ramírez-Castañeda, Diogo Veríssimo.

**Formal analysis:** Tatsuya Amano.

**Funding acquisition:** Tatsuya Amano, Shawan Chowdhury.

**Investigation:** Tatsuya Amano, Violeta Berdejo-Espinola, Israel Borokini, Shawan Chowdhury, Marina Golivets, Juan David González-Trujillo, Flavia Montaño-Centellas, Kumar Paudel, Rachel Louise White.

**Methodology:** Tatsuya Amano, Valeria Ramírez-Castañeda, Diogo Veríssimo.

**Project administration:** Tatsuya Amano, Violeta Berdejo-Espinola.

**Validation:** Tatsuya Amano, Violeta Berdejo-Espinola.

**Visualization:** Tatsuya Amano.

**Writing – original draft:** Tatsuya Amano.

**Writing – review & editing:** Tatsuya Amano, Valeria Ramírez-Castañeda, Violeta Berdejo-Espinola, Israel Borokini, Shawan Chowdhury, Marina Golivets, Juan David González-Trujillo, Flavia Montaño-Centellas, Kumar Paudel, Rachel Louise White, Diogo Veríssimo.

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
