## [Editor Report · Decision Letter 0]

19 Dec 2022

Dear Tatsuya, 

Thank you for submitting your manuscript entitled "The cost of being a non-native English speaker in science" for consideration as a Research Article by PLOS Biology.

Your manuscript has now been evaluated by the PLOS Biology editorial staff, as well as by an academic editor with relevant expertise, and I'm writing to let you know that we would like to send your submission out for external peer review.

IMPORTANT: Please change the article type to "Meta-Research Article" when you upload your additional metadata (see next paragraph).

Once your full submission is complete, your paper will undergo a series of checks in preparation for peer review. After your manuscript has passed the checks it will be sent out for review. To provide the metadata for your submission, please Login to Editorial Manager (https://www.editorialmanager.com/pbiology) within two working days, i.e. by Dec 21 2022 11:59PM.

Kind regards,

Roli

Roland Roberts, PhD

Senior Editor

PLOS Biology

rroberts@plos.org

---

## [Decision Letter · Decision Letter 1]

27 Feb 2023

Dear Tatsuya,

Thank you for your patience while your manuscript "The cost of being a non-native English speaker in science" went through peer-review at PLOS Biology. Your manuscript has now been evaluated by the PLOS Biology editors, an Academic Editor with relevant expertise, and by three independent reviewers. Please accept my apologies for the delay incurred over the holiday period during the early stages of the editorial process.

You’ll see that all three reviewers are positive about your study, but each raises a number of issues that should be addressed before further consideration. For example, reviewer #1 wants you to spell out the limitation that it focuses on fields that may be atypical in terms of collaboration patterns, reviewer #2 raises a number of interesting issues that can probably be addressed textually, and reviewer #3 suggests that you make better use of your Figs by binning all the “>50 papers” data points together, thereby presenting the bulk of the data at higher resolution. In addition, two reviewers request that you incorporate more extensive discussion of potential solutions to the problems identified; both suggest moving Table S18 into the main paper. 

In light of the reviews, which you will find at the end of this email, we are pleased to offer you the opportunity to address the comments from the reviewers in a revision that we anticipate should not take you very long. We will then assess your revised manuscript and your response to the reviewers' comments with our Academic Editor aiming to avoid further rounds of peer-review, although might need to consult with the reviewers, depending on the nature of the revisions.

**IMPORTANT - SUBMITTING YOUR REVISION**

*Resubmission Checklist*

*Published Peer Review*

*PLOS Data Policy*

*Blot and Gel Data Policy*

Sincerely,

Roli

Roland Roberts, PhD

Senior Editor

PLOS Biology

rroberts@plos.org

REVIEWERS' COMMENTS:

Reviewer #1:

[identifies himself as Scott Edmunds]

This is the review of "The cost of being a non-native English speaker in science" by Tatsuya Amano and colleagues. This tackles a very important issue that is a major impediment to the effectiveness and productivity of scientific systems, holding back global participation in the research process.

I am reviewing this as someone working in the publishing industry for 15 years, and an Open Science advocate with experience and a perspective living in non-English speaking countries (E and SE Asia) for most of my career. I do not have the statistical expertise to assess those parts, and have also not carried out survey-based research, so hope that other referees are able to assess these areas sufficiently. 

A strength of the study is that it is trying to look purely at the cost of the language barriers, ignoring many of the other geographic biases in science that may exist. Although these are hard confounders to remove. For example as conferences and conference papers are covered here, many of the nationalities covered will have financial and immigration issues attending conferences in North America and Europe that are arguably much more challenging and stressful than just the language issues. And even online conferences may not be completely immune to some of these inclusivity issues (see and overview here https://doi.org/10.1093/gigascience/giab051). So one potential challenge for this study is whether all of these bigger picture societal issues have been properly taken into account in the study design and discussion.

The topic in question - quantifying costs and drawbacks of English being the common language of science is extremely timely and topical with nearly 200 member states ratifying in late-2021 the UNESCO Recommendation of Open Science. Making a commitment to apply the provisions of these by taking whatever legislative or other measures may be required to give effect within their jurisdictions. UNESCO spend a lot of time in the Recommendations outlining the core values and guiding principles of Open Science, and among these "Equity and fairness", "Diversity and inclusiveness", "Equality of opportunities", and "Collaboration, participation and inclusion" all highlight language as a key issue.

The UNESCO report ends with areas of action and 8 specific recommendations, one of these being "Encouraging multilingualism in the practice of science, in scientific publications and in academic communications." To address such important changes data is required, and member states are also asked to monitor uptake and the benefits of open science as a result of the implementation of these recommendations. Providing quantitative data with regards to one of the key areas highlighted is therefore a useful outcome of this work, and it would be useful to highlight that topicality in the introduction and discussion. 

See: UNESCO Recommendation on Open Science, November 2021. https://unesdoc.unesco.org/ark:/48223/pf0000379949.locale=en

A major limitation not highlighted sufficiently is the fact that this is being sold as a study quantifying these issues in all of science, and when looking at the methodology and sample collected this data is very specific to a few particular areas of science. Namely conservation, ecology and evolutionary biology. While this fits with the "Biology" scope of PLOS Biology, this major limitation definitely needs to be made much clearer at the start of the introduction, probably in the abstract, and maybe even the title. The language issue is likely to be relatively generalisable, but without more data it is impossible to say if these figures are the same across all fields of science. Where some fields may be more theoretical, mathematical or computational. With more lab-based (financial) resource heavy fields skewed to researchers with bigger international collaboration networks providing them better access to expensive equipment and reagent, and English speaking co-authors and collaborators to ask favours from. Or if there are fields with a longer history of undergraduate teaching using English language textbooks, or that are so fast moving only English language teaching resources exist. 

Where these types of meta-research studies can be very successful is where they don't just quantify a problem, but also propose some potential solutions too. See the Ioannidis PLOS Medicine paper "How to Make More Published Research True" as an example of this (https://doi.org/10.1371/journal.pmed.1001747). In the case of this work, there is little to no discussion on this in the main paper, and the main mention of this is hidden in the supplementary files table S18. I would strongly make the case to make this more prominent as a table or infographic in the discussion section of the main paper. Especially as S18 isn't supplementary data, it is an important part of the narrative.

Some other minor points that could be useful to clarify and think about. 

Looking at figure 1 charts on Language barriers in paper reading and writing, it looks like the total number of English-language papers published is less for the non-native English speaking authors and that career limitation is sad in itself. In Fig 1A it looks like for readers with moderate English-proficiency, once they get beyond a certain level of seniority they are slightly faster at reading English papers than native English readers. Is that just an artifact or can it have another explanation? 

While researchers in low and lower middle income countries will find it hard to afford expensive editing services, I'm curious if you have an idea why this group of researchers also were less frequent in asking someone to help as a favour? Do you feel this is likely because of a lack of access to contacts that can help or another reason?

Machine translation is dismissed as not being relevant due to low usage rate in the countries surveyed. The usefulness of this is likely to vary based on languages that have better trained and working translation tools such as Chinese (and may explain the slightly higher rate in Japanese). With the recent release of ChatGPT AI/NLP guided or enhanced writing is an extremely fast moving and topical area, and while writing created by AI is being prohibited by universities and journals, there has been less discussion on AI-utilising tools that more subtly improve and polish writing such as Grammarly. This was missed off the survey but it would have been interesting to know the uptake of tools such as these in these different research communities. And could potentially be a topic for follow-up.

Reviewer #2:

[identifies herself as Jo Havemann, Access 2 Perspectives]

Referring to the manuscript draft with the title: The cost of being a non-native English speaker in science (PBIOLOGY-D-22-02711R1 )

First, thank you for investigating this urgent topic by quantifying the actual cost to career opportunities of translation in research. It is shocking, even for people like me who are already sensitized to the issue in certain disciplines and editorial decision makings - to see numbers to the effect that language proficiency has on career-defining activities in academia. You make a compelling case highlighting the cost of translation to non-native English-speaking researchers from various national backgrounds with varying English proficiency and national income levels as based on international standards. 

You contextualize the issue with the urgency for political action re biodiversity and climate change, where we have a dire need for a research-informed global society by tapping into various knowledge systems a.o. 

Methodology

The number of individuals successfully recruited for the survey with 40 to 294 per linguistic/economic groups appears adequate to allow for the conclusions drawn, even though you argue for a relatively low sample size yourselves. The translation efforts are well-categorized reflecting the real-world scenarios in engaging in the foreign (secondary, tertiary, …) language for scientific purposes. 

With the design of the survey, allowing for adequate self-assessment in responding to the quantifying questions, as well as survey distribution in each of the target countries' official language, you made appropriate efforts to invite for informed participation of a large enough group in each country. 

Data anonymization was well taken care of as well as informing participants of the voluntary nature of the survey.

Discussion

The style chosen in the first paragraph of the discussion is compelling, where the reader is invited to put him/her/them-self/selves into the shoes of a non-native English speaking ECR, which makes the challenge even more immediate and approachable. 

You present sound examples and conclusions about biases in the career development of non-native English speakers being selected against based on their English-language proficiency (line 217-220).

You further argue for economic disadvantages being the reason for hindrances in utilizing English language editing services, and what effect the presence or absence of scientific terms and glossaries in certain language group may have on tertiary educational choices amongst the population of a country (line 221-231). 

The limitations of your study are coherently addressed with underestimated disadvantages during the planning phase that you became aware of during the survey analysis.

Recommendations

Not to ignore research communities that are communicating in their local and other large regional languages - such as Arabic, French, Spanish, Chinese, and Russian, you could perhaps set a scope for your appeal for those disciplines and research communities or venues where English is the preferred or only accepted language for journal article publishing.

In line 87 you mention reading as "a requisite for obtaining necessary knowledge in research". Please keep in mind scientific literature that is published in languages other than English in local and regional journals and repositories.

Terms such as "Global North" and "Global South" carry substantial weight in assumptions for economic versus geographical attributes and might be misleading or inappropriate especially in the context of this article. I would suggest to instead be naming countries directly or referring to specific geographical regions and continents and/or populations of particular language groups. 

In line 40 and 47 you mention "under-represented communities/groups". Could you please specify what you mean by that and in reference to whom? The argument seems to go beyond the distinction between native and non-native English speakers. 

While it is important to highlight disadvantages, the manuscript could benefit from mentioning more explicitly the advantages of multilingualism in a globalized world and advocating for multilingual research practices to be adopted by all researchers, non-native English speakers and native English speakers alike.

What is striking to me is that non-native English speakers spend less time in preparing presentations or manuscripts in their first language as compared to native English speakers, in each of their native language (line 176ff). Were you able to identify explanations for this?

Additional comments

Thank you for also specifying author contribution according to the CRediT taxonomy.

From my personal experience and that of many of my colleagues, it could also be inquired and highlighted how non-native English speakers lose the skill to explain their work in their native language, once they fully emerged in the English language to conduct and communicate about their research in and what effect that might have on science literacy in any part of the world irrespective of economic capacity; perhaps to consider in follow-up investigations and discussions.

Reviewer #3:

This article describes the various disadvantages in the current international scientific system faced by scientists who are not native English speakers. The results are not surprising: it takes non-native English speakers longer to read articles and to write articles, and their articles are more likely to be rejected. And non-native English speakers tend to avoid English-language conferences and take more time to prepare talks. But while the results are not surprising, the paper provides quantitative information on a problem that is pervasive and generally known in the field. 

The abstract should mention that the people surveyed are in the field of environmental science.

The sample sizes are not large, but they are reasonable and adequate. 

The results are presented in a clear way that makes the problem easy to understand. The quantitative results are easy to see, though the authors may want to reduce the number of decimal places being reported, 46.64 is probably one too many decimal places. 

Figure 1 is not very effective. For example, let's consider the first item of the time needed to read a paper. Most of the right side of the figure is empty as the number of people writing more than 50 papers is very small. Each of these figures need to be redone, perhaps by lumping together all of the people with more than 50 papers into one group. 

In the Discussion, the point about survivor bias is very good. The people who have remained in the field and continued to publish are likely the people who have had better English language skills from the start, who have learned English over time, or have figured out how to get help, like language services or collaborators. The authors also make the point that even within a country, there is a variability in language skills and opportunities; even in a poor country where will be some well-off families with excellent education opportunities for their children. 

An important part of improving the current system would be to suggest solutions. Some of these are provided in the Discussion, but some of the more specific suggestions are only in Table S18 where most readers will not see them. The authors should try to bring this table into the main paper and mention some of the points in the Discussion. In particular, the authors might consider mentioning the idea of "buddies".

The journal Biological Conservation has had various article in recent years that consider this same topic, but from a different perspective and a different methodology. The authors should consult these papers, and if appropriate, incorporate them into their paper. 

Maas B, RJ Pakeman, L Godet, L Smith, V Devictor, RB Primack. 2021. Women and Global South strikingly underrepresented among top-publishing ecologists. Conservation Letters 14 (4), e12797

This study found a striking under-representation of top-publishing authors from non-English speaking countries

Primack, R.B. and L. Zipf. 2016. Editorial: Acceptance rates and number of papers in Biological Conservation from 2005 to 2014 for Australia, Brazil, China, India, Spain, and the United States: Trends or noise? Biological Conservation 196:50-52. 

There is a low acceptance rate of papers from China and other countries, but the acceptance rate for China is increasing over time. 

Primack, R.B., Marrs, R., 2008. Bias in the review process. Biological Conservation 

141, 2919-2920.

There is a strong disadvantage in getting papers accepted for non-English speakers, including even for high-income countries like France and Germany.

---

## [Editor Report · Decision Letter 2]

18 May 2023

Dear Tatsuya,

Thank you for your patience while we considered your revised manuscript "The cost of being a non-native English speaker in science" for publication as a Meta-Research Article at PLOS Biology. This revised version of your manuscript has been evaluated by the PLOS Biology editors and the Academic Editor.

Based on our Academic Editor's assessment of your revision, we are likely to accept this manuscript for publication, provided you satisfactorily address the following data and other policy-related requests.

a) While the nature of the costs of being a non-native English speaker are nuanced, it's clear that these are multiple and diverse, so we suggest that you make the title more emphatic, as follows: "The manifold costs of being a non-native English speaker in science"

c) Please address my Data Policy requests below; specifically, we need you to supply the numerical values directly underlying Figs 1ABCDEF, 2ABCD, 3AB, 4ABC, S1, S2, S3, S4, S5, S6, S7, S8, S9, either as a supplementary data file or as a permanent DOI’d deposition. We note that you have already deposited code and some data in OSF; please clarify the relationship between this and the Figures in this paper (I do understand that the raw survey data are protected by the agreement with the ethics office, and are therefore exempt from PLOS’ data policy).

d) Please cite the location of the data clearly in all relevant main and supplementary Figure legends, e.g. “The data underlying this Figure can be found in S1 Data” or “The data underlying this Figure can be found in https://doi.org/10.5281/zenodo.XXXXX”
https://osf.io/XXXXX

We expect to receive your revised manuscript within two weeks. 

*Published Peer Review History*

*Press*

Sincerely,

Roli

Roland Roberts, PhD

Senior Editor,

rroberts@plos.org,

PLOS Biology

DATA POLICY:

Regardless of the method selected, please ensure that you provide the individual numerical values that underlie the summary data displayed in the following figure panels as they are essential for readers to assess your analysis and to reproduce it: Figs 1ABCDEF, 2ABCD, 3AB, 4ABC, S1, S2, S3, S4, S5, S6, S7, S8, S9. NOTE: the numerical data provided should include all replicates AND the way in which the plotted mean and errors were derived (it should not present only the mean/average values).

DATA NOT SHOWN?

---

## [Editor Report · Decision Letter 3]

5 Jun 2023

Dear Tatsuya,

Thank you for the submission of your revised Meta-Research Article "The manifold costs of being a non-native English speaker in science" for publication in PLOS Biology. On behalf of my colleagues and the Academic Editor, Ulrich Dirnagl, I'm pleased to say that we can in principle accept your manuscript for publication, provided you address any remaining formatting and reporting issues. These will be detailed in an email you should receive within 2-3 business days from our colleagues in the journal operations team; no action is required from you until then. Please note that we will not be able to formally accept your manuscript and schedule it for publication until you have completed any requested changes.

I'm sorry about the problems that you had in confirming your compliance with our data policy requirements while I was on vacation and Nonia was at a conference. It seems that you were able to reach a satisfactory conclusion with her in my absence, so all's well that ends well!

Sincerely,

Roli

Senior Editor

PLOS Biology

rroberts@plos.org